# LEO: GENERATIVE LATENT IMAGE ANIMATOR FOR HUMAN VIDEO SYNTHESIS

## ABSTRACT

Spatio-temporal coherency is a major challenge in synthesizing high quality videos, particularly in synthesizing human videos that contain rich global and local deformations. To resolve this challenge, previous approaches have resorted to different features in the generation process aimed at representing appearance and motion. However, in the absence of strict mechanisms to guarantee such disentanglement, a separation of motion from appearance has remained challenging, resulting in spatial distortions and temporal jittering that break the spatio-temporal coherency. Motivated by this, we here propose LEO, a novel framework for human video synthesis, placing emphasis on spatio-temporal coherency. Our key idea is to represent motion as a sequence of flow maps in the generation process, which inherently isolate motion from appearance. We implement this idea via a flow-based image animator and a Latent Motion Diffusion Model (LMDM). The former bridges a space of motion codes with the space of flow maps, and synthesizes video frames in a warp-and-inpaint manner. LMDM learns to capture motion prior in the training data by synthesizing sequences of motion codes. Extensive quantitative and qualitative analysis suggests that LEO significantly improves coherent synthesis of human videos over previous methods on the datasets TaichiHD, FaceForensics and CelebV-HQ. In addition, the effective disentanglement of appearance and motion in LEO allows for two additional tasks, namely infinite-length human video synthesis, as well as content-preserving video editing.

## 1 INTRODUCTION

Deep generative models such as generative adversarial networks (GANs) (Goodfellow et al., 2014) and Diffusion Models (Ho et al., 2020; Song et al., 2021) have fostered a breakthrough in video synthesis (Vondrick et al., 2016; Tulyakov et al., 2018; Saito et al., 2020; Wang et al., 2020b; 2021b; Yu et al., 2022; Skorokhodov et al., 2022; Ge et al., 2022; Singer et al., 2023; Villegas et al., 2023; Ho et al., 2022a), elevating tasks such as text-to-video generation (Singer et al., 2023; Villegas et al., 2023), video editing (Bar-Tal et al., 2022), as well as 3D-aware video generation (Bergman et al., 2022). While existing work has demonstrated promising results w.r.t. frame-wise visual quality, synthesizing videos of strong spatio-temporal coherency, tailored to human videos, containing rich global and local deformations, remains challenging.

Motivated by this, we here propose an effective generative framework, placing emphasis on *spatio-temporal coherency* in *human video synthesis*. Having this in mind, a fundamental step has to do with the *disentanglement* of videos w.r.t. *appearance* and *motion*. Previous approaches have tackled such disentanglement by two jointly trained distinct networks, respectively providing appearance and motion features (Tulyakov et al., 2018; Wang et al., 2020b; 2021b; 2020a; Yu et al., 2022), as well as by a two-phase generation pipeline that firstly aims at training an image generator, and then at training a temporal network to generate videos in the image generator's latent space (Tian et al., 2021; Ge et al., 2022; Yan et al., 2021). Nevertheless, such approaches encompass limitations related to spatial artifacts (*e.g.*, distortions of body structures and facial identities in the same sequence), as well as temporal artifacts (*e.g.*, inter-frame semantic jittering), even in short generated videos of 16 frames. We argue that such limitations stem from incomplete disentanglement of appearance and motion in the generation process. Specifically, without predominant mechanisms or hard constraints to guarantee disentanglement, even a minor perturbation in the high-level semantics will be amplified and will lead to significant changes in the pixel space.

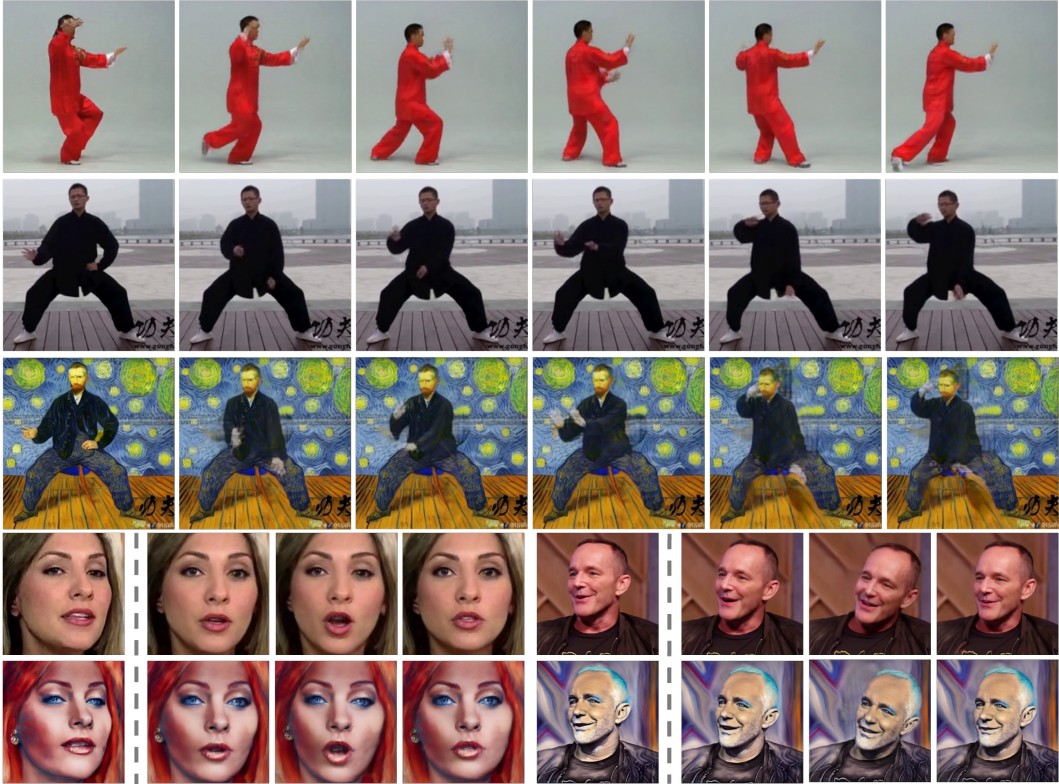

Figure 1: Our framework caters a set of video synthesis tasks including (i) unconditional video generation (first and second row), (ii) conditional generation based on one single image (fourth row) and (iii) video editing from the starting image (third and fifth row). Results pertain to our model being trained on the datasets TaichiHD, FaceForensics and CelebV-HQ.

Deviating from the above and towards disentangling videos w.r.t. appearance and motion, in this paper we propose a novel framework for human video generation, referred to as LEO, streamlined to ensure strong *spatio-temporal coherency*. At the core of this framework is a *sequence of flow maps*, representing *motion semantics*, which inherently isolate motion from appearance. Specifically, LEO incorporates a latent motion diffusion module (LMDM), as well as a flow-based image animator. In order to synthesize a video, an initial frame is either provided externally for *conditional generation*, or obtained by a generative module for *unconditional generation*. Given such initial frame and a sequence of motion codes sampled from the LMDM, the flow-based image animator generates a sequence of flow maps, and proceeds to synthesize the corresponding sequence of frames in a *warp-and-inpaint* manner.

The *training* of LEO is decomposed into *two phases*. *Firstly*, we train the flow-based image animator to encode input images into low-dimensional latent motion codes, and map such codes to flow maps, which are used for reconstruction via warp-and-inpaint. Therefore, once trained, the flow-based image animator naturally provides a space of motion codes that are strictly constrained to only containing motion-related information. At the *second stage*, upon the space provided by the image animator, we train the LMDM to synthesize sequences of motion codes and capture *motion prior* in the training data. To endow LEO with the ability to synthesize videos of arbitrary length beyond the short training videos, we adopt a Linear Motion Condition (LMC) mechanism in LMDM. As opposed to directly synthesizing sequences of motion codes, LMC enables LMDM to synthesize sequences of residuals w.r.t. a starting motion code, in order for longer videos to be easily obtained by concatenating additional sequences of residuals.

To evaluate LEO, we conduct extensive experiments pertained to three human video datasets, including TaichiHD (Siarohin et al., 2019), FaceForensics (Rössler et al., 2018), and CelebV-HQ (Zhu et al., 2022). Compared to previous video synthesis methods, LEO demonstrates a significantly im-

proved spatio-temporal coherency, even on synthesized videos of length of 512 frames. In addition, LEO shows great potential in two extended tasks, namely *infinite-length video synthesis*, as well as *video editing* of a style in a synthesized video, while maintaining the content of the original video.

## 2 RELATED WORKS

**Unconditional video generation** aims to generate videos by learning the full distribution of training dataset. Most of the previous works (Vondrick et al., 2016; Saito et al., 2017; Tulyakov et al., 2018; Wang et al., 2020b; Wang, 2021; Wang et al., 2021b; Clark et al., 2019; Brooks et al., 2022) are built upon GANs (Goodfellow et al., 2016; Radford et al., 2015; Brock et al., 2019; Karras et al., 2019; 2020) towards benefiting from the strong performance of the image generator. Approaches (Denton & Birodkar, 2017; Li & Mandt, 2018; Bhagat et al., 2020; Xie et al., 2020) based on VAEs (Kingma & Welling, 2014) were also proposed while only show results on toy datasets. Recently, with the progress of deep generative models (*e.g.*, VQVAE (Van Den Oord et al., 2017), VQGAN (Esser et al., 2021), GPT (Radford et al., 2018) and Denoising Diffusion Models (Ho et al., 2020; Song et al., 2021; Nichol & Dhariwal, 2021)) on both image (Ramesh et al., 2021; 2022) and language synthesis (Radford et al., 2019), as well as the usage of large-scale pre-trained models, video generation also started to be explored with various approaches.

MoCoGANHD (Tian et al., 2021) builds the model on top of a well-trained StyleGAN2 (Karras et al., 2020) by integrating an LSTM in the latent space towards disentangling content and motion. DIGAN (Yu et al., 2022) and StyleGAN-V (Skorokhodov et al., 2022) and MoStGAN-V (Shen et al., 2023), inspired by NeRF (Feichtenhofer et al., 2019), proposed an implicit neural representation approach to model time as a continuous signal aiming for long-term video generation. VideoGPT (Yan et al., 2021) and TATS (Ge et al., 2022) introduced to first train 3D-VQ models to learn discrete spatio-temporal codebooks, which are then be refined temporally by modified transformers (Vaswani et al., 2017). Recently, VDM (Ho et al., 2022b) has shown promising capacity to model complex video distribution by incorporating spatio-temporal operations in Diffusion Models. While previous approaches have proposed various attempts either in training strategies (Tian et al., 2021; Yan et al., 2021; Ge et al., 2022) or in model architectures (Wang et al., 2020b; 2021b; Yu et al., 2022; Skorokhodov et al., 2022) to disentangle appearance and motion, due to the lack of strong constrains, it is still difficult to obtain satisfying results.

In contrast to unconditional video generation, **conditional video generation** seeks to produces high-quality videos, following image-to-image generation pipeline (Chu et al., 2017; Isola et al., 2017; Huang et al., 2018). In this context, additional signals such as semantic maps (Pan et al., 2019; Wang et al., 2018; 2019), human key-points (Jang et al., 2018; Yang et al., 2018; Walker et al., 2017; Chan et al., 2019; Zakharov et al., 2019; Wang et al., 2019; 2021a), motion labels (Wang et al., 2020a), 3DMM (Zhao et al., 2018; Yang et al., 2022) and optical flow (Li et al., 2018; Ohnishi et al., 2018) have been exploited to guide motion generation. In addition, text description, has been used in large-scale video diffusion models (Singer et al., 2023; Ho et al., 2022a; Blattmann et al., 2023; Wang et al., 2023; Ho et al., 2022b) for high-quality video generation. Our framework also supports for conditional video generation based on a single image. However, unlike previous approaches, our method follows the image animation pipeline (Siarohin et al., 2019; 2021; Wang et al., 2022) which leverages the dense flow maps for motion modeling. We introduce our method in details in the following.

## 3 METHOD

Fig. 2 illustrates the training of LEO, comprising of two-phases. We firstly train an image animator towards learning high-quality latent motion codes of the datasets. In the second phase, we train the Latent Motion Diffusion Model (LMDM) to learn a motion prior over the latent motion codes. To synthesize a video, the pre-trained image animator takes the motion codes to generate corresponding flow maps, which are used to warp and inpaint starting frame to produce the video sequence.

We formulate a video sequence $v = \{x_i\}_{i=1}^{L}, x_i \sim \mathcal{X} \in \mathbb{R}^{3 \times H \times W}$ as $v = \{\mathcal{T}(x_1, G(\alpha_i))\}_{i=2}^{L}, \alpha_i \sim \mathcal{A} \in \mathbb{R}^{1 \times N}$, where $x_i$ denotes the $i^{th}$ frame, $\alpha_i$ denotes a latent motion code at timestep $i$, $G$ represents the generator in the image animator aiming to generate a flow map $\phi_i$ from $\alpha_i$. The generated frame is obtained by warping $(\mathcal{T})$ $x_1$ using $\phi_i$.

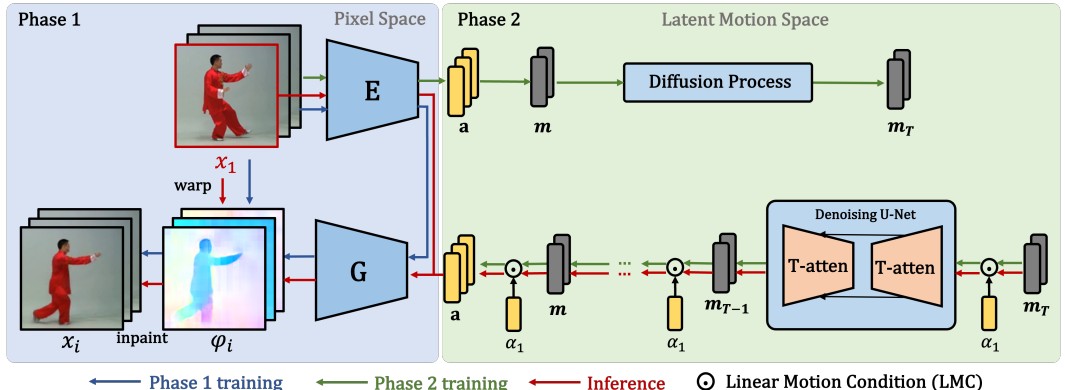

Figure 2: **Overview of LEO.** Our framework incorporates two main parts, (i) an image animator, aiming to generate flow maps and synthesize videos in the pixel space, and (ii) Latent Motion Diffusion Model (LMDM), focusing on modeling the motion distribution in a latent motion space. Our framework requires a two-phase training. In the first phase, we train the image animator in a self-supervised manner towards mapping latent codes to corresponding flow maps $\phi_i$. Once the image animator is well-trained, motion codes $\mathbf{a}$ are extracted from a frozen encoder and used as inputs of LMDM. In the second phase, LMDMs are trained to learn the motion distribution by providing the starting motion $\alpha_1$ as condition. Instead of directly learning the distribution of $\mathbf{a}$, we adopt a Linear Motion Condition (LMC) mechanism in LMDM towards synthesizing sequences of residuals with respect to $x_1$. At the inference stage, given a starting image $x_i$ and corresponding motion code $\alpha_i$, LMDM firstly generates a motion code sequence, which is then used by the image animator to generate flow maps to synthesize output videos in a warp-and-inpaint manner.

### 3.1 LEARNING LATENT MOTION CODES

Towards learning a frame-wise latent motion code, we adopt the state-of-the-art image animator LIA (Wang et al., 2022) as it enables to encode input images into corresponding motion codes. LIA consists of two modules, an encoder $E$ and a generator $G$. During training, given a source image $x_s$ and a driving image $x_d$, $E$ encodes $x_s, x_d$ into a motion code $\alpha = E(x_s, x_d)$, and $G$ generates a flow field $\phi = G(\alpha)$ from the code. LIA is trained in a self-supervised manner with the objective to reconstruct the driving image.

Training LIA in such a self-supervised manner brings two notable benefits for our framework, (i) it enables LIA to achieve high-quality perceptual results, and (ii) as a motion code is strictly equivalent to flow maps, there are guarantees that $\alpha$ is only motion-related without any appearance interference.

### 3.2 LEANING A MOTION PRIOR

Once LIA is well-trained on a target dataset, for any given video $v = \{x_i\}_{i=1}^{L}$, we are able to obtain a motion sequence $\mathbf{a} = \{\alpha_i\}_{i=1}^{L}$ with the frozen $E$. In the second phase of our training, we propose to learn a motion prior by temporal Diffusion Models.

Unlike image synthesis, data in our second phase is a set of sequences. We firstly apply a temporal Diffusion Model for modeling the temporal correlation of $\mathbf{a}$. The general architecture of this model is a 1D U-Net adopted from (Ho et al., 2020). To train this model, we follow the standard training strategy with a simple mean-squared loss,

$$L_{\text{LMDM}} = \mathbb{E}_{\epsilon \sim \mathcal{N}(0,1), t} \left[ \| \epsilon - \epsilon_\theta(\mathbf{a}_t, t) \|_2^2 \right], \tag{1}$$

where $\epsilon$ denotes the unscaled noise, $t$ is the time step, $\mathbf{a}_t$ is the latent noised motion code to time $t$. During inference, a random Gaussian noise $\mathbf{a}_T$ is iteratively denoised to $\mathbf{a}_0 = \{\alpha_i\}_{i=1}^{L}$, and the final video sequence is obtained through the generator.

At the same time in our experiments, we found that learning motion sequences in a complete unconditional manner brings to the fore limitations, namely (i) the generated codes are not consistent

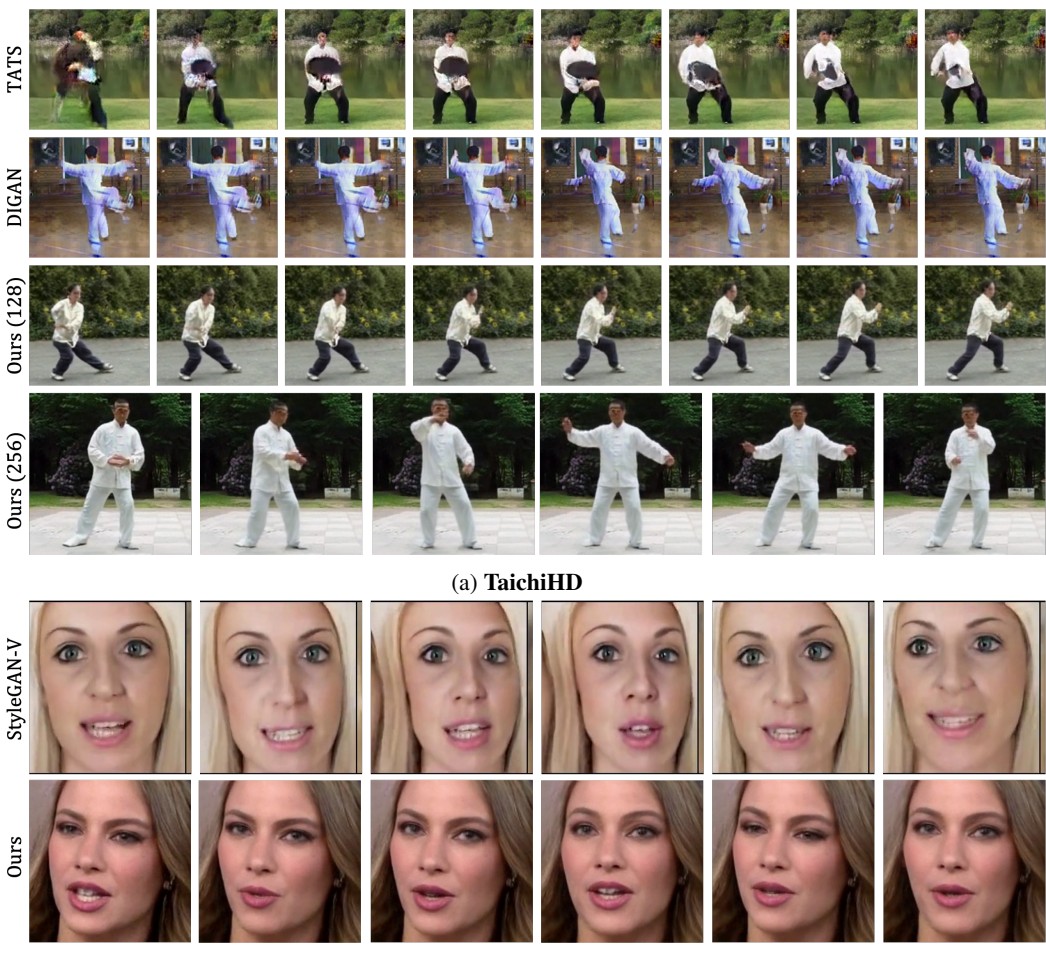

Figure 3: **Qualitative Comparison.** We qualitatively compare LEO with DIGAN, TATS, StyleGAN-V on short video generation. The results indicate that on both (a) TaichiHD (128 and 256 resolutions) and (b) FaceForensics datasets, our proposed method achieves the best visual quality and is able to capture the human structure well. Other approaches either modify the facial structure (e.g., StyleGAN-V) or fail to generate a complete human body (e.g., TATS and DIGAN).

enough for producing smooth videos, as well as (ii) the generated motion codes can only be used to produce fixed length videos. Hence, towards addressing those issues, we propose a **conditional Latent Motion Diffusion Model (cLMDM)** which aims for high-quality and long-term human videos.

One major characteristic of LIA has to do with the linear motion space. Any motion code $\alpha_t$ in $\mathbf{a}$ can be re-formulated as

$$\alpha_i = \alpha_1 + m_i, i \geq 2, \tag{2}$$

where $\alpha_1$ denotes the motion code at the first timestep and $m_i$ denotes the motion difference between timestep 1 and $i$, so that we can re-formulate $\mathbf{a}$ as

$$\mathbf{a} = \alpha_1 + \mathbf{m}, \tag{3}$$

where $\mathbf{m} = \{m_i\}_{i=2}^{L}$ denotes the motion difference sequence. Therefore, Eq. 2 and 3 indicate that a motion sequence can be represented by $\alpha_1$ and $\mathbf{m}$. Based on this, we propose a **Linear Motion Condition (LMC)** mechanism in cLMDM to condition the generative process with $\alpha_1$. During training, at each time step, we only add noise onto $\mathbf{m_t}$ instead of the entire $\mathbf{a}$ and leave $\alpha_1$ intact. The objective function of cLMDM is

$$L_{\text{cLMDM}} = \mathbb{E}_{\epsilon \sim \mathcal{N}(0,1),t}\left[\|\epsilon - \epsilon_\theta(\mathbf{m}_t, \alpha_1, t)\|_2^2\right], \tag{4}$$

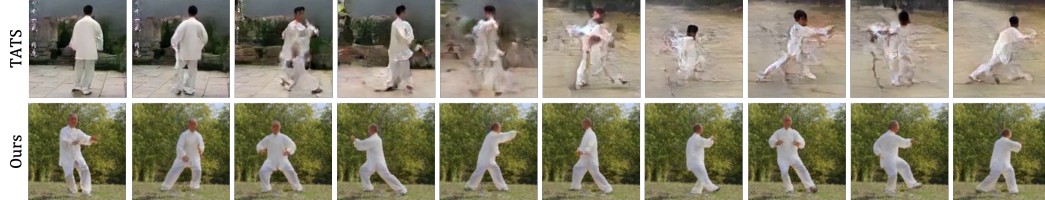

Figure 4: **Comparison on long-term video generation.** We compare with TATS by generating 512-frame videos. Videos from TATS start crashing around 50 frames while our model is able to continue producing high-quality frames with diverse motion.

where $\alpha_1$ denotes the condition signal and $\mathbf{m}_t$ stands for the noised $\mathbf{m}$ to time $t$. $\alpha_1$ is first added on $\mathbf{m}_t$ and then concatenated along temporal dimension. LMC will be applied at each time step until we reach $\mathbf{m}_0$. The final motion sequence is obtained as $\mathbf{a}_0 = [\alpha_1, \mathbf{m}_0]$. We find that following this, a related generated motion sequence is more stable and contains fewer artifacts, as $\alpha_1$ serves as a strong signal to constrain the generated $\mathbf{m}$ to follow the initial motion.

While the results from cLMDM outperforms previous models, the groundtruth $\alpha_1$ is necessitated during both, training and inference stage. Towards *unconditional generation*, we train an additional simple DM to fit the distribution $p(\alpha_i)$ in a frame-wise manner. We refer to the cLMDM and such simple DM jointly as **Latent Motion Diffusion Model (LMDM)**. By this way, LMDM are able to work in both conditional and unconditional motion generation.

Towards generating videos of arbitrary length, we propose an autoregressive approach based on proposed LMDM. By taking the last motion code from the previous generated sequence as the $\alpha_1$ in the current sequence, with a randomly sampled noise, LMDM are able to generate an infinite-length motion sequence. By combining such sequence in pre-trained LIA with a starting image, LEO can synthesize photo-realistic and long-term videos.

### 3.3 LEARNING STARTING FRAMES

In our framework, a starting image $x_1$ is required to synthesize a video. As image space is modeled independently, here we propose two options to obtain $x_1$.

**Option 1: existing images.** The first option is to directly take the images either from a real distribution or from an image generation network. In this context, our model is a conditional video generation model, which learns to predict future motion from $x_1$. Starting motion $\alpha_1$ is obtained through $\alpha_1 = E(x_1)$.

**Option 2: conditional Diffusion Models.** The second option is to learn a conditional DDPM (Ho et al., 2020) (cDDPM) with $\alpha_1$ as a condition to synthesize $x_1$. By combining LEO with LMDM as well as cDDPM, we are able to conduct unconditional video synthesis.

## 4 EXPERIMENTS

In this section, we firstly briefly describe our experimental setup, introducing datasets, evaluation metrics and implementation details. Secondly, we qualitatively demonstrate generated results on both, short and long video synthesis. Then we show quantitative evaluation w.r.t. video quality, comparing LEO with SoTA. Next, we conduct an ablation study to prove the effectiveness of proposed conditional mechanism LMC. Finally, we provide additional analysis of our framework, exhibiting motion and appearance disentanglement, video editing and infinite-length video generation.

**Datasets.** As we focus on human video synthesis, evaluation results are reported on three human-related datasets, TaichiHD (Siarohin et al., 2019), FaceForensics (Rössler et al., 2018) and CelebV-HQ (Zhu et al., 2022). We use both $128 \times 128$ and $256 \times 256$ resolution TaichiHD datasets, and only $256 \times 256$ resolution FaceForensics and CelebV-HQ datasets.

**Evaluation metric.** For quantitative evaluation, we apply the commonly used metrics FVD and KVD, in order to compare with other approaches on video quality and apply Average Content Dis-

| Method | TaichiHD128 | | | TaichiHD256 | | FaceForensics | | CelebV-HQ |
| | $FVD_{16}$ | $KVD_{16}$ | $ACD_{16}$ | $FVD_{16}$ | $KVD_{16}$ | $FVD_{16}$ | $ACD_{16}$ | $FVD_{16}$ |
|---|---|---|---|---|---|---|---|---|
| MoCoGAN-HD | $144.7 \pm 6.0$ | $25.4 \pm 1.9$ | - | - | - | 111.8 | 0.33 | 212.4 |
| DIGAN | $128.1 \pm 4.9$ | $20.6 \pm 1.1$ | 2.17 | $156.7 \pm 6.2$ | - | 62.5 | - | 72.9 |
| TATS | $136.5 \pm 1.2^*$ | $22.2 \pm 1.0^*$ | 2.28 | - | - | - | - | - |
| StyleGAN-V | - | - | - | - | - | 47.4 | 0.36 | 69.1 |
| MoStGAN-V | - | - | - | - | - | 39.7 | 0.38 | 132.1 |
| Ours (uncond) | $100.4 \pm 3.1$ | $11.4 \pm 3.2$ | 1.83 | $122.7 \pm 1.1$ | $20.49 \pm 0.9$ | 52.3 | 0.28 | - |
| Ours (cond) | $\mathbf{57.6 \pm 2.0}$ | $\mathbf{4.0 \pm 1.5}$ | $\mathbf{1.22}$ | $\mathbf{94.8 \pm 4.2}$ | $\mathbf{13.47 \pm 2.3}$ | $\mathbf{35.9}$ | $\mathbf{0.27}$ | $\mathbf{40.2}$ |

Table 1: **Evaluation for unconditional and conditional short video generation.** LEO systematically outperforms other approaches on conditional video generation, and achieves better or competitive results on unconditional generation w.r.t. FVD, KVD and ACD. (*results are reproduced based on official code and released checkpoints.)

| Method | TaichiHD128 | | | FaceForensics | |
| | $FVD_{128}$ | $KVD_{128}$ | $ACD_{128}$ | $FVD_{128}$ | $ACD_{128}$ |
|---|---|---|---|---|---|
| DIGAN | - | - | - | 1824.7 | - |
| TATS | $1194.58 \pm 1.1$ | $462.03 \pm 8.2$ | 2.85 | - | - |
| StyleGAN-V | - | - | - | $\mathbf{89.34}$ | 0.49 |
| Ours | $\mathbf{155.54 \pm 2.6}$ | $\mathbf{48.82 \pm 5.9}$ | $\mathbf{2.06}$ | 96.28 | $\mathbf{0.34}$ |

Table 2: **Evaluation for unconditional long-term video generation.** LEO outperforms other methods on long-term (128 frames) video generation w.r.t. FVD, KVD and ACD.

tance (ACD) towards evaluating the identity consistency of faces and bodies in the generated videos. In addition, we conduct a user study with 20 users towards comparing with objective quantitative evaluation.

**Implementation details.** We implement LEO using PyTorch (Paszke et al., 2019). The architecture of the LIA and LMDM are adopted from (Wang et al., 2022) and (Nichol & Dhariwal, 2021), respectively. Both LIA and LMDM are trained using 4 NVIDIA A100 GPUs.

More details about datasets, evaluation metrics, implementation and training details are described in App. A.1.

### 4.1 QUALITATIVE EVALUATION

We qualitatively compare LEO with SoTA by visualizing the generated results. We firstly compare our method with DIGAN, TATS and StyleGAN-V on the FaceForensics and TaichiHD datasets for *short video generation*. As shown in Fig. 1 and 3, the visual quality of our generated results outperforms other approaches w.r.t both, appearance and motion. For both resolutions on TaichiHD datasets, our method is able to generate complete human structures, whereas both, DIGAN and TATS fail, especially for arms and legs. When compared with StyleGAN-V on FaceForensics dataset, we identify that while LEO preserves well facial structures, StyleGAN-V modifies such attributes when synthesizing large motion.

Secondly, we compare with TATS for long-term video generation. Specifically, 512 frames are produced for the resolution $128 \times 128$ pertained to the TaichiHD dataset. As shown in Fig. 4, the subject in the videos from TATS starts crashing around 50 frames and the entire video sequence starts to fade. On the other hand, in our results, the subject continues to perform diverse actions whilst well preserving the human structure. We note that our model is only trained using a 64-frame sequence.

### 4.2 QUANTITATIVE EVALUATION

In this section, we compare our framework with five SoTA methods for both, conditional and unconditional short video generation, as well as unconditional long-term video generation.

**Unconditional short video generation.** In this context, as described in Sec. 3.3, Option 2, the $x_1$ is randomly generated by a pre-trained cDDPM. We compare with SoTA by generating 16 frames.

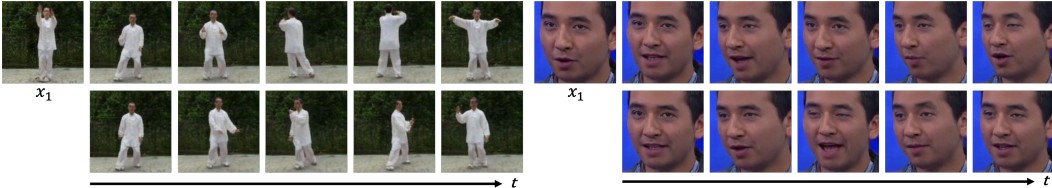

Figure 5: **Disentanglement of motion and appearance.** The first and second row share the same appearance, with different motion codes. Results display that our model is able to produce diverse motion from the same content.

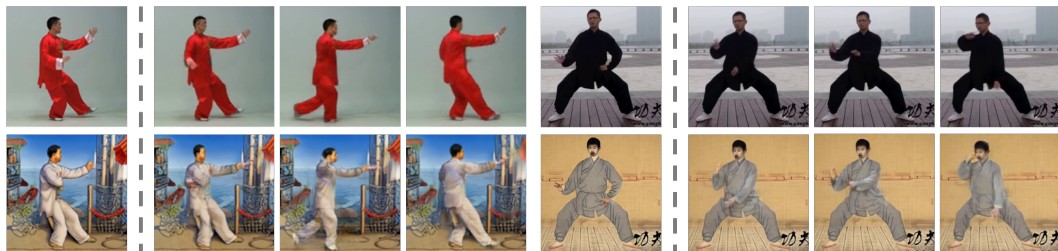

Figure 6: **Video editing.** We show video editing results by combining LEO with off-the-shelf image editing model ControlNet. We are able to edit the appearance of the entire video sequence through only editing the starting image.

To compare with DIGAN on high-resolution generation, we also generate videos of $256 \times 256$ resolution. Related FVDs and KVDs are reported in Tab. 1. LEO systematically outperforms other methods w.r.t. video quality, obtaining lower or competitive FVD and KVD on all datasets. On high-resolution generation, our results remain better than DIGAN.

However, by comparing the results between StyleGAN-V and ours, we find FVD is not able to represent the quality of generated videos veritably. We observe that StyleGAN-V is not able to preserve facial structures, whereas LEO is able to do so, see Fig. 3. We additionally compute ACD, in order to further analyze the identity consistency in 16-frame videos. Tab. 1 reflects on the fact that our method achieves significantly better results compared to other approaches. In addition, the user study confirms this, as nearly all users rated for our generated results to be more realistic. Hence, we conclude that a metric, replacing FVD is in urgent need in the context of video synthesis.

**Unconditional long video generation** We evaluate our approach for long-term video generation w.r.t. FVD and ACD. In this context, we compare LEO with StyleGAN-V on the FaceForensics dataset, and both DIGAN and TATS on the TaichiHD. We report results based on 128-frame generation in Tab. 2, which clearly shows that our method outperforms others in such context. We hypothesize that consistent and stable motion codes produced by our LMDM are key to producing high-quality long-term videos.

**Conditional short video generation** As described in Sec. 3.3, Option 1, our framework additionally caters for conditional video generation by taking an existing image to hallucinate the following motion. Specifically, we randomly select 2048 images from both, TaichiHD and FaceForensics datasets as $x_1$ and compute corresponding $\alpha_1$ as input of LMDM. As depicted in Tab. 1, results conditioned on the real images achieve the lowest FVD, KVD and ACD values, suggesting that the quality of a starting image is pertinent for output video quality, which further signifies that in the setting of unconditional generation, training a better cDDPM will be instrumental for improving results.

## 5 ABLATION STUDY

In this section, we place emphasis on analyzing the effectiveness of proposed Linear Motion Condition (LMC) in LMDM. We train two models, with and without LMC on both TaichiHD and Face-

| Method | TaichiHD (%) | FaceForensics (%) |
|---|---|---|
| Ours / TATS | **93.00** / 7.00 | - |
| Ours / StyleGAN-V | - | **91.33** / 8.67 |

Table 3: **User study.** We conduct a user study pertaining to the datasets TaichiHD and FaceForensics which show that our results are more realistic than TATS and StyleGAN-V.

| | TaichiHD | FaceForensics |
|---|---|---|
| w/o LMC | 118.6 | 60.03 |
| with LMC | **100.4** | **52.32** |

Table 4: **Ablation study of proposed LMC.** Models with LMC achieved the lowest FVD on both datasets.

Forensics datasets. As shown in Tab. 4, using LMC significantly improves the generated video quality, which proves that our proposed LMC is an effective mechanism for involving $\alpha_1$ in LMDM.

## 6 ADDITIONAL ANALYSIS

**Motion and appearance disentanglement.** We proceed to combine the same $x_1$ with different $\mathbf{m}$, aiming to reveal whether $\mathbf{m}$ is only motion-related. Fig. 5 illustrates that different $\mathbf{m}$ enables the same subject to perform different motion - which proves that our proposed LMDM is indeed learning a motion space, and appearance and motion are clearly disentangled. This experiment additionally indicates that our model does not overfit on the training dataset, as different noise sequences are able to produce diverse motion sequences.

**Video Editing.** As appearance is modeled in $x_1$, we here explore the task of video editing by modifying the semantics in thestarting image. Compared to previous approaches, where image-to-image translation is required, our framework simply needs an edit of the semantics in an one-shot manner. Associated results are depicted in Fig. 1 and Fig. 6. We apply the open-source approach ControlNet (Zhang & Agrawala, 2023) on the starting frame by entering various different prompts. Given that the motion space is fully disentangled from the appearance space, our videos maintain the original temporal consistency, uniquely altering the appearance.

**Infinite-length video generation.** Towards exploring the limits of our framework, we explore infinite-length video generation using our LMDM. Infinite-length video generation is highly challenging, as the model is required to retain producing semantically meaningful motion codes, systematically matching the content in a starting images. We firstly study the autoregressive approach to recurrently apply our LMDM. Surprisingly, we find that such a simple approach is sufficient to produce more than 1000 frames on the FaceForensics dataset. Various motion patterns can be generated by randomly sampling noises. However, for the Taichi dataset, it appears that subjects repeat previous actions. By analyzing the results, we infer the problem has to do with the limited motion patterns occurring in Taichi. Once the motion sequence fails in certain type of pattern, it becomes challenging for transitions to others. Towards solving this problem, we build a simple *transition diffusion model* to help the network exit the current loop. In this way, we are able to generate infinite-length Taichi with diverse motion patterns. Details are provided in the App. A.3.

In addition, we discuss limitations of our approach in App. A.4.

## 7 CONCLUSIONS

In this paper, we introduced LEO, a novel framework incorporating a Latent Image Animator (LIA), as well as a Latent Motion Diffusion Model (LMDM), placing emphasis on spatio-temporal coherency in human video synthesis. By jointly exploiting LIA and LMDM in a two-phase training strategy, we endow LEO with the ability to disentangle appearance and motion. We quantitatively and qualitatively evaluated proposed method on both, human body and talking head datasets and demonstrated that our approach is able to successfully produce photo-realistic, long human videos. In addition, we showcased that the effective disentanglement of appearance and motion in LEO allows for two additional tasks, namely infinite-length human video synthesis by autoregressively applying LMDM, as well as content-preserving video editing (employing an off-the-shelf image editor (e.r., ControlNet)). We postulate that LEO opens a new door in design of generative models for video synthesis and plan to extend our method onto more general videos and applications.

## ETHIC STATEMENT

In this work, we aim to synthesize high-quality human-centric videos by combining a pretrained image animator with a proposed latent motion diffusion model. Our approach can be used for digital human, online education, and data synthesis for other computer vision tasks, etc. We note that our framework mainly focuses on learning how to model motion distribution in a pretrained image animator rather than directly model appearance. Therefore, our framework is not biased towards any specific gender, race, region, or social class. It works equally well irrespective of the difference in subjects.

## REPRODUCIBILITY STATEMENT

We assure that all the results shown in the paper and supplemental materials can be reproduced. We intend to open-source our code, as well as trained models.

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

# A APPENDIX

We here describe our detailed experimental setup in App. A.1 including datasets, evaluation metrics, as well as implementation details. Then in App. A.3, we introduce the setting of infinite-length video generation, pertaining to both datasets, FaceForensics and TaichiHD.

## A.1 EXPERIMENTS

### A.1.1 DATASETS

**TaichiHD** (Siarohin et al., 2019) comprises 3100 video sequences downloaded from YouTube. In train and test splits, it contains 2815 and 285 videos, respectively. We conducted all our experiments on the train split and used both $128 \times 128$ and $256 \times 256$ resolutions in our experiments.

**FaceForensics** (Rössler et al., 2018) includes 1000 video sequences downloaded from YouTube. Following the preprocessing of previous methods (Saito et al., 2020; Skorokhodov et al., 2022), face areas are cropped based on the provided per-frame meshes. We resized all videos to $256 \times 256$ resolution.

**CelebV-HQ** (Zhu et al., 2022) comprises 35666 high-quality talking head videos of 3 to 20 seconds each. In total, it represents 15653 celebrities. We resized the original videos to $256 \times 256$ resolution, in order to train our models.

### A.1.2 EVALUATION METRICS

**Frechet video distance (FVD) and Kernel Video Distance (KVD).** We use I3D (Carreira & Zisserman, 2017) trained on Kinetics-400 as feature extractor to compute FVD and KVD. However, we find FVD is a very sensitive metric, which can be affected by many factors such as frame-rate, single image quality, video length and implementation, which also mentioned in (Skorokhodov et al., 2022). Therefore, towards making a fair comparison, on the TaichiHD dataset, we adopt the implementation from DIGAN (Yu et al., 2022). As for FaceForensics and CelebV-HQ, we chose to follow the implementation of StyleGAN-V (Skorokhodov et al., 2022).

**Average Content Distance (ACD).** ACD measures the content consistency in generated videos. To evaluate results from FaceForensics and TaichiHD, we extract features from each generated frame and proceed to extract a per-frame feature vector in a video. The ACD was then computed using the average pairwise L2 distance of the per-frame feature vectors. We follow the implementation in (Tian et al., 2021) to compute ACD for FaceForensics. As for TaichiHD, we employ the pre-trained person-reID model (Zheng et al., 2018) to extract person identity features.

**User study.** We asked 20 human raters to evaluate generated video quality. Towards evaluating quality, we show paired videos and ask the raters, to rate 'which clip is more realistic'. Each video-pair contains one generated video from our method, whereas the second video is either *real* or generated from other methods.

### A.1.3 IMPLEMENTATION DETAILS

Our framework requires two-phase training. In the first phase, we follow the standard protocol to train LIA (Wang et al., 2022) to encode input images into low-dimensional latent motion codes, and map such codes to flow maps, which are used for reconstruction via warp-and-inpaint. Therefore, once trained, LIA naturally provides a space of motion codes that are strictly constrained to only containing motion-related information. In the second phase, we only train LMDM on the extracted motion codes from Encoder. We note that the LMDM is a 1D U-Net adopted from (Nichol & Dhariwal, 2021), we set the input size as $64 \times 20$, where 64 is the length of the sequence and 20 is the dimension of the motion code. We use 1000 diffusion steps and a learning rate of $1e-4$. As the training of LMDM is conducted in the latent space of LIA, the entire training is very efficient and only requires one single GPU.

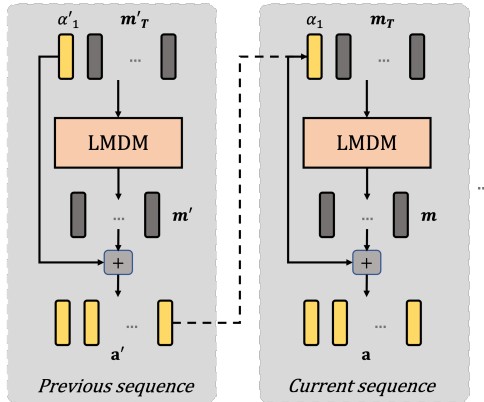

Figure 7: Infinite-length video generation for FaceForensics.

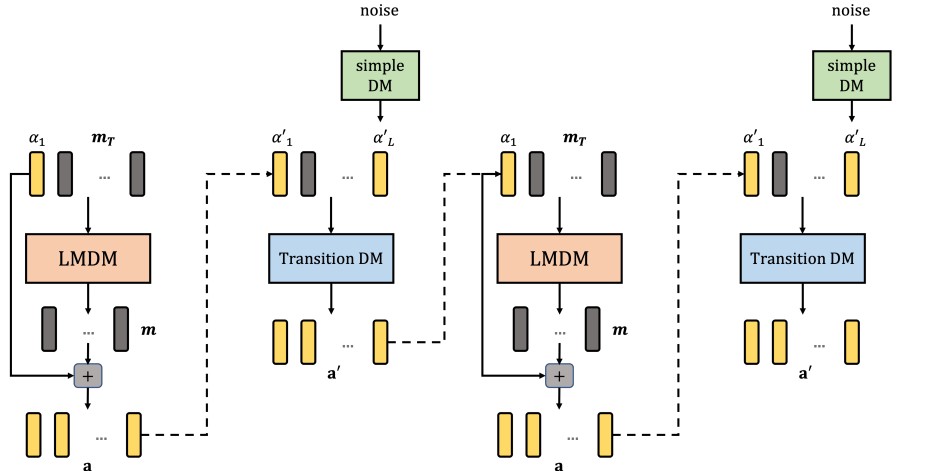

Figure 8: Infinite-length video generation for TaichiHD.

## A.2 GENERAL VIDEO GENERATION

We conduct unconditional video generation on UCF101, results are shown in Tab. 5

| Methods | $FVD_{16}$ |
|---|---|
| MoCoGAN-HD | 1729.6 |
| DIGAN | 1630.2 |
| StyleGAN-V | 1431.0 |
| MoStGAN-V | 1380.3 |
| Ours | 1256.2 |

Table 5: Quantitative evluation on UCF101 *w.r.t.* FVD.

## A.3 INFINITE-LENGTH VIDEO GENERATION

In addition to presented settings, our framework is able to generate infinite-length videos. To generate long-term FaceForensics, as shown in Fig. 7, we provide the last generated code from the previous sequence as the starting code of the current sequence. The entire long-term video is generated in an *autoregressive* manner. We note that for TaichiHD dataset, due to limited motion patterns, this setting yields repeated motion. Towards addressing this limitation, as shown in Fig. 8, we design an additional *transition diffusion model (DM)* aimed at generating transition motion between

the last code from original generated sequence and a new motion code generated from the *simple DM*. Doing so, the transition DM enforces the network to exit the original motion pattern and transit to new pattern.

## A.4 LIMITATIONS AND FUTURE WORK

We list several limitations in current framework and proposed potential solutions for future work.

- *Geometry ambiguity and temporal coherency.* Since we use a 2D generator to predict 2D flow maps, LEO is not able to handle human body occlusion very well especially in Taichi dataset. One solution would be to incorporate the architecture of NeRF or Tri-plane into our generator to support 3D-aware generation. We think in this way, the issues of geometry ambiguity and human body occlusion could be addressed.

- *Generalizability.* Since the pre-trained image animator focuses on talking head and human bodies, our proposed framework currently only achieves good performance on human-centric videos. However, we think it is the limitation of current pretrained image animator rather than the proposed framework LEO. Once the image animator is improved for general video data, our framework can be easily adapted for general image animation and video generation, such as text-to-video generation.

- *Architecture.* Current architect of LEO still relies on convolutional networks in both image animator and latent motion diffusion models. Advanced techniques such as transformers have not been explored yet. Future work would be involving novel architecture design and training LEO on larger-scale dataset to explore the limits of current approach.

