# OpenReview forum: "LEO: Generative Latent Image Animator for Human Video Synthesis"
_ICLR.cc/2024/Conference — Submitted to ICLR 2024_

### Official Review · Reviewer_NikU · 2023-10-29

**Soundness:** 3 good
**Presentation:** 3 good
**Contribution:** 2 fair
**Rating:** 3
**Confidence:** 4

**Summary:**

This work introduces a diffusion-based method for video generation. The proposed method leverages a flow-based image animator to learn motion representations thus enabling disentangle motion from appearance. An LDM is designed to learn the motion distribution by providing the starting motion α1 as the condition.

**Strengths:**

1. This work tries to solve the challenging issue of disentangling motion from appearance. The method is well-motivated and the proposal method is simple to understand.
2. A Linear Motion Condition (LMC) mechanism is designed in cLMDM to condition the generative process with the first motion code α1.
3. Qualitative results show the ability to generate long videos and enable disentanglement of motion and appearance.

**Weaknesses:**

1. The author only includes pickup methods for comparison, STOA methods are not included for comparison. Recent methods, such as MoStGAN-V, VDM, Video-LDM, VideoFactory, and Make-A-Video, should be included for comparison.

2. The author should include experiments on more challenging datasets, such as MSR-VTT and UCF101.

**Questions:**

see weakness.

---

> ### Author Response · Authors · 2023-11-20
> **Response from the Authors**
>
> We thank the reviewer for the feedback, stating that (1) our proposed method is well-motivated and simple, (2) our proposed idea is effective for appearance and motion disentanglment, as well as long video generation.
>
> **Q1. The author only includes pickup methods for comparison, STOA methods are not included for comparison. Recent methods, such as MoStGAN-V, VDM, Video-LDM, VideoFactory, and Make-A-Video, should be included for comparison.**
>
> **A1.** We have included listed works in Related Works in the revised manuscript.
>
> For MostGAN-V,  we have updated Tab. 1 in revised manuscript and Supplementary Material website on both FaceForensics and CelebV datasets. Results show that our method outperforms MoStGAN-V both qualitatively and quantitatively.
>
> For VDM, Video-LDM, VideoFactory, and Make-A-Video, **none of the listed methods is officially open-sourced, as well as reported any evaluation results on FaceForensics, CelebV and Taichi datasets.** Qualitativley and quantitatively comaprison with those methods would be difficult.
>
> We note that our work differs from listed methods in the tasks to solve. We focus on human-centric video generation, while the listed methods mainly target to solve general unconditional video or text-to-video generation problems.  Technically, the listed methods aim to generate very short video clips (e.g., 16 frames before interpolation) to align with text input. They do not propose any techniques for appearance and motion disentanglement, nor for long video generation, which, however, are the main targets in our work.  We think our proposed techniques could be a complementery component in general video generation system.
>
> **Q2. The author should include experiments on more challenging datasets, such as MSR-VTT and UCF101**
>
> **A2.** From our point of view, articulated human video generation is a very challenging task, even large-scale diffusion models can not handle it very well. It requires the model to simultaneously learn the human structure and motion in both spatial and temporal dimensions.
>
> We have provided quantitative evaluation on UCF101 for unconditonal generation in Appendix A.2 Tab. 5 in revised manuscript. Results show that our method outperforms state-of-the-art w.r.t. FVD. However, from the results, we observe that it is still challenging for LEO in general video generation. We have addressed this limitation in the revised manuscript in Appendix A.4 and will leave it for our future work.
>
> We argue that MSR-VTT (text-video dataset) and UCF101 (label-video dataset) are well-known for general conditional video generation, especially for label-/text-to-video generation. We think they are not proper datasets to evaluate the contributions of our method as LEO is designed for human-centric data rather than general video data. We focus on appearance and motion disentanglment, as well as long video generation. We have quantitatively and qualitatively evaluated LEO on three widely used human-centric datasets.

---

> > ### Comment · Reviewer_NikU · 2023-11-21
> >
> > In the response, the author claims that the work mainly focuses on human-centric video generation, while the listed methods mainly target solving general unconditional video or text-to-video generation problems. Actually, I do not see much difference between human-centric video generation and general video generation. The challenge of ensuring spatiotemporal coherency lies not only in human-centric video generation but also in general video generation.
> >
> > Second, I do not think it's a proper evaluation without real STOA included for comparison. For example, on the TaiChi dataset, LVDM and VideoFusion should be compared. Moreover, VideoFusion reports better performance than the proposed method on this dataset.
> >
> > He Y, Yang T, Zhang Y, et al. Latent video diffusion models for high-fidelity long video generation[J]. arXiv preprint arXiv:2211.13221, 2023.
> > Luo Z, Chen D, Zhang Y, et al. VideoFusion: Decomposed Diffusion Models for High-Quality Video Generation[C]//Proceedings of the IEEE/CVF Conference on Computer Vision and Pattern Recognition. 2023: 10209-10218.

---

> > > ### Author Response · Authors · 2023-11-23
> > > **Response from the Authors (part 1/2)**
> > >
> > > We thank Reviewer NiKU for the quick feedback, and provide our comments as follows:
> > >
> > > **A1.** We have a different perspective on this. While ensuring spatiotemporal coherency is the target for video generation, in different video generation tasks, the requirements of spatiotemporal coherency are different.
> > >
> > > Despite the remarkable success achieved by current text-to-video (T2V) systems, it is important to highlight their primary focus on generating very short videos that lack complex motion. While state-of-the-art systems, such as Runway, demonstrate the potential to produce text-aligned spatial content, the resulting movements tend to be relatively simplistic. The majority of motion observed in these videos primarily involves global motion, such as camera movements (e.g., zoom in or zoom out), or simple local motion of specific subjects (e.g., pouring coffee, swimming shark, etc.). **While these results may suffice for generating simple dynamic scenes, they fall short of meeting the requirements for achieving spatio-temporal coherence in complex human-centric videos.**
> > >
> > > The main challenge that persists across all T2V systems lies in generating actions involving articulated structures, particularly in the context of humans. For instance, producing a 1-minute video of a single person performing TaiChi remains a significant hurdle. We conjecture that the lack of explicit or implicit modeling of human structure within current T2V systems may be the underlying cause of this challenge. However, the optimal approach to incorporate such modeling into the T2V pipeline remains an open question, deserving further exploration.
> > >
> > > Therefore, in our work, we proposed an implicit approach for modeling human motion. We do not rely on any explicit structure representations (e.g., skeletons) but learn a set of motion codes to reprent motion distribution. In this way, the entire system is completely data-driven and can be learned in a self-supervised manner. Thanks to such design, motion and appearance can be well-disentangled, and long video generation can be achieved by only modeling motion code in latent space. Results demonstrate that spatio-temporal cohenrency can be better achieved compared to state-of-the-art.
> > >
> > > We consider our proposed approach to be a potential complementery work compred to the general video generation system and could provide potential insight for future works to design more advanced video generation system.

---

> > > > ### Author Response · Authors · 2023-11-23
> > > > **Response from the Authors (part 2/2)**
> > > >
> > > > **A2.**
> > > > **Comparison to LVDM:** LVDM proposed a masking mechanism to autoregressively apply diffusion models for long video generation. However, different from our approach, they do not disentangle appearance and motion, nor leverage flow maps.
> > > >
> > > > We achieved lower FVD (ours: 94.8 vs LVDM: 99.0) on Taichi dataset. We also qualitatively show the generated Taichi results in the updated supplementary material website. Results show that our generated videos have better spatio-temporal consistency and contain larger and more complex motion. Since we did not find any released checkpoints for long video generation on the Taichi dataset, comparison on this task might be difficult.
> > > >
> > > > We would like to kindly remind reviewer that LVDM is only an arxiv paper and has not been peer-reviewed yet.
> > > >
> > > > **Comparison with VideoFusion:** **Unfortunetely, we did not managed to find the opensource code, as well as checkpoints on Tachi dataset.** Hence, fair comparison would be difficult. However, we test their online text-to-video platform via https://huggingface.co/spaces/damo-vilab/modelscope-text-to-video-synthesis with the prompt "a man playing Taichi".
> > > > Generated videos are shown in the updated supplementary material website. From the results we observe that LEO achieves better visual quality and smoother motion compared to VideoFusion, which shows that human action generation is still challenging for current T2V system.
> > > >
> > > > As pointed by Reviewer 8SWB, VideoFusion leverages the prior knowledge in a pretrained DALLE for video generation, which might help it to achieve lower FVD. We agree this point, and would like to note that FVD is not a precise evaluation metric to measure video quality, qualitative results are still important for fair comparison.
> > > >
> > > > In general, VideoFusion decomposes each frame in a video into a base and residual part, which represented by a base noise, as well as a residual noise in the spatio-temporal joint latent space. Base noise is shared across time dimensions, and residual noises are time individual. Base noise is added on to residual noise to represent each frame. Both base and residual noises are still 2D matrices, which represent both spatial and temporal information.
> > > >
> > > > Different from VideoFusion, LEO decomposes the motion into a starting motion, as well as a sequence of residual motion codes in the latent space. Our motion codes are 1D vectors, hence are easily to be modeled by a seq2seq model. The starting motion is represented by a ground truth/generated motion code rather than noise. Our experiments demonstrate that appearance and motion disentanglemtn in pixel space, and decomposing motion in latent space help generate better visual quality, as well as longer videos.
> > > >
> > > > We hope our answers have addressed your concerns. If you have any remaining questions or concerns, we are happy to address them. Thank you for your time and consideration!

---

### Official Review · Reviewer_8SWB · 2023-10-29

**Soundness:** 3 good
**Presentation:** 3 good
**Contribution:** 3 good
**Rating:** 6
**Confidence:** 3

**Summary:**

The paper presents a temporal generative model LEO for synthesizing editable human performance video. The key idea is to represent motions with optical flow maps to disentangle appearances and dynamics. In particular, LEO leverages a latent diffusion model trained for predicting motions in an auto-regressive fashion, and decodes the latent to form flow maps for pixel-space appearance synthesis.

Their quantitative results show obvious improvement over prior work, with qualitative evidence demonstrating better spatio-temporal coherency.

**Strengths:**

The paper is well-written and easy to follow, and the proposed solution sounds solid. Particularly:
- Novel formulation of diffusion-based generative model for optical flow generations, which enables long-term motion generation
- Explicit disentanglement of the video into appearance (pixel values) and motion (optical flow) that makes LEO better preserve the identity information in the input.
- Auto-regressive motion generation with careful designs that achieve long-term video generation.

The quantitative and qualitative evaluations also show significant improvement over prior arts.

**Weaknesses:**

While showing promising results, LEO has some limitations, which are also observed in other baselines:
- Geometry ambiguity: without any explicit notion of 3D geometry or semantic features, LEO often flips or morphs the limbs from one side to the other. This is particularly obvious in the TaichiHD videos.
- Temporal coherency: while LEO improves greatly over the other baselines compared in the paper, the appearance can still drift off/morph arbitrarily between frames, especially for videos with occlusion/dis-occlusions or large motions.
- Limitations and failure cases: these aspects are not presented in the papers and supplementary. Proper discussions on what LEO cannot do well can help the readers to better assess the contribution of the work, and also open up possible future directions.

**Questions:**

Below are the questions I have:
- How does the proposed LEO compare to the approaches like Siarohin et al. 2019; 2021, where the motions are disentangled into region-based descriptors/flow-field?
- What are the limitations of LEO? What are the failure cases? It would be great if the paper could show and discuss these topics.

---

> ### Author Response · Authors · 2023-11-20
> **Response from the Authors**
>
> We thank the reviewer for the positive feedback, and appreciate that reviewer finds (1) our formulation is novel, (2) our appearance and motion disengagement better preserves identity, and (3) our carefully designed method achieves long video generation.
>
> **Q1. How does the proposed LEO compare to the approaches like Siarohin et al. 2019; 2021, where the motions are disentangled into region-based descriptors/flow-field?**
>
> **A1.** Firstly, Siarohin et al. 2019; 2021 proposed leveraging predicted explicit structure representations, i.e., 2D keypoints and region descriptors to generate flow maps. However, such 2D structural representations could not be well-predicted in a self-supervised learning manner. Regions and keypoints are unable to be precisely predicted for large motion, which leads to deformation in faicial structures. Instead, motion code in our method is a per-frame 1D vector. It is able to capture both local and global motion, as well as preserve the faical structure even if in long video generation. We show comparison to Siarohin et al. 2019; 2021 on image animation in updated Supplementary Material website.
>
> Secondly, as region-based descriptor in Siarohin et al. 2021 is a 2D representation. It is unclear how to use such representation to build motion models for unconditional video generation, as well as long video generation. While in our work, motion code can be easily modeled by a standard seq2seq diffusion model.
>
> **Q2. What are the limitations of LEO? What are the failure cases? It would be great if the paper could show and discuss these topics.**
>
> **A2.** In the revised manuscript, we have addressed the limitations of our approach in Appendix A.4. Additionally, we have provided updated failure cases in the Supplementary Material website for further analysis. It is worth noting that, due to the autoregressive nature of our method for long video generation, there are instances where the model may inadvertently repeat the same action in videos of considerable length. To mitigate this issue, we propose a potential solution in the form of introducing random tiny perturbations to the motion code. By incorporating these subtle variations, we aim to enhance diversity and avoid repetitive patterns in the generated videos.

---

> > ### Comment · Reviewer_8SWB · 2023-11-22
> >
> > I would like to thank the authors for addressing my questions.
> >
> > Reviewer NikU raised the question of insufficient quantitative comparisons against other concurrent video diffusion models, such as VideoFusion and LVDM. In my opinion, this is a valid concern, and comparisons to VideoFusion are possible, given that they also provide results on Taichi. LEO does not necessarily need to perform better than VideoFusion, given how VideoFusion uses a pre-trained image decoder (DALL-E) in their method, but it would be great to see proper comparisons and analysis against it.

---

> > > ### Author Response · Authors · 2023-11-23
> > > **Response from the Authors**
> > >
> > > We thank Reviewer 8SWB for the quick feedback, and provide our comments as follows:
> > >
> > > **Comparison to LVDM:** LVDM proposed a masking mechanism to autoregressively apply diffusion models for long video generation. However, different from our approach, they do not disentangle appearance and motion, nor leverage flow maps.
> > >
> > > We achieved lower FVD (ours: 94.8 vs LVDM: 99.0) on Taichi dataset. We also qualitatively show the generated Taichi results in the updated supplementary material website. Results show that our generated videos have better spatio-temporal consistency and contain larger and more complex motion. Since we did not find any released checkpoints for long video generation for Taichi dataset, comparison on this task might be difficult.
> > >
> > > We would like to kindly remind reviewer that LVDM is only an arxiv paper and has not been peer-reviewed yet.
> > >
> > > **Comparison to VideoFusion:** **Unfortunetely, we did not managed to find the opensource code, as well as checkpoints on Tachi dataset.** Hence, fair comparison would be difficult. However, we tested their online text-to-video platform via https://huggingface.co/spaces/damo-vilab/modelscope-text-to-video-synthesis with the prompt "a man playing Taichi".
> > > Generated videos are shown in the updated supplementary material website. From the results we observe that LEO achieves better visual quality and smoother motion compared to VideoFusion, which shows that human action generation is still challenging for current T2V system.
> > >
> > > In general, VideoFusion decomposes each frame in a video into a base and residual part, which represented by a base noise, as well as a residual noise in the spatio-temporal joint latent space. Base noise is shared across time dimensions, and residual noises are time individual. Base noise is added on to residual noise to represent each frame. Both base and residual noises are still 2D matrices, which represent both spatial and temporal information.
> > >
> > > Different from VideoFusion, LEO decomposes the motion into a starting motion, as well as a sequence of residual motion codes in the latent space. Our motion codes are 1D vectors, hence are easily to be modeled by a seq2seq model. The starting motion is represented by a ground truth/generated motion code rather than noise. Our experiments demonstrate that appearance and motion disentanglemtn in pixel space, and decomposing motion in latent space help generate better visual quality, as well as longer videos.
> > >
> > > We hope our answers have addressed your concerns. If you have any remaining questions or concerns, we are happy to address them. Thank you for your time and consideration!

---

> > > > ### Comment · Reviewer_8SWB · 2023-11-23
> > > >
> > > > Thanks for the detailed analysis and quick responses, I appreciate it.
> > > >
> > > > I agree with the authors that comparisons with LVDM are neither possible nor necessary as it has not yet been published. The analysis of VideoFusion also seems fair to me, especially considering that (1) LEO and VideoFusion are not designed for the same task (motion-to-video versus text-to-video), and (2) VideoFusion leverages a pre-trained decoder. I think the quantitative comparisons, as they stand now, are already sufficient.

---

### Official Review · Reviewer_H4YQ · 2023-11-01

**Soundness:** 3 good
**Presentation:** 3 good
**Contribution:** 3 good
**Rating:** 6
**Confidence:** 4

**Summary:**

The paper proposes a method to generate videos by disentangling the synthesis of appearance and motion. To this end, the authors propose a flow-based image animator and a latent motion diffusion model. In particular, the motion synthesis is conditioned on the starting motion code. This formulation allows the model to generate sequences of infinite length by changing the starting frame for each subsequence. The efficacy of the method is evaluated on multiple datasets of humans in motion.

**Strengths:**

- The application of synthesizing videos of arbitrary length is relevant and challenging.
- The main idea is simple and clearly presented.
- The quantitative and qualitative results showcase the efficacy of the proposed model over the baselines on the TaichiHD, FaceForensics and CelebV-HQ datasets.

**Weaknesses:**

- It would be nice to see some human-specific baselines, especially since the focus of the paper is on humans, e.g., utilizing skeleton/3DMM guidance.
- I believe a comparison (or at least discussion) to video-ldm [1] would be beneficial.
- I am missing a section on the limitations and ethical considerations.

[1] Align your Latents: High-Resolution Video Synthesis with Latent Diffusion Models

**Questions:**

In general, I am positively inclined however I would suggest that the authors address the issues raised in the "weaknesses" section, especially regarding the human-specific baselines and the limitations/ethical considerations.

---

> ### Author Response · Authors · 2023-11-20
> **Response from the Authors**
>
> We thank the reviewer for the positive feedback, in particular that (1) our task is challenging, (2) our main idea is simple and clearly presented, and (3) our quantitative and qualitative results showcase the efficacy of the method.
>
> **Q1. It would be nice to see some human-specific baselines, especially since the focus of paper is on humans, e.g., utilizing skeleton/3DMM guidance.**
>
> **A1.**  We compared our method with a 3DMM-based talking head generation method [1], as well as two diffusion-based conditional generation methods using skeleton as guidance [2, 3]. We showed several results on the Taichi and FaceForensics datasets in the updated Supplementary Material website.
>
> Upon analysis, we observed that the results of the diffusion-based methods [2, 3] were significantly influenced by the quality of the extracted poses and skeletons. In contrast, our proposed method exhibited superior visual quality and temporal coherency. Furthermore, when compared to the 3DMM-based method [1], our approach achieved notable improvements in terms of facial details and head motion.
>
> [1] Min et al., StyleTalk: One-shot Talking Head Generation with Controllable Speaking Styles, AAAI 2023
>
> [2] Zhang et al, ControlVideo: Training-free Controllable Text-to-Video Generation, arXiv:2305.13077
>
> [3] Khachatryan et al., Text2Video-Zero: Text-to-Image Diffusion Models are Zero-Shot Video Generators, ICCV 2023
>
> **Q2. I believe a comparison or at least discussion to video-ldm would be beneficial.**
>
> **A2.** Video-LDM is a large-scale video generation framework, which extends a pretrained Stable Diffusion with temporal modules in both VAE and 2D UNet. Video features are first extracted by VAE-Encoder,  and then used in spatio-temporal UNet to learn the latent video distribution. In Video-LDM, appearance and motion are not disentangled, and spatio-temporal distribution is jointly learned in 3D UNet. In addition, Video-LDM is only able to generate 16-frame short video clips.
>
> In our work, we pre-trained an image animator on human face and body. LIA is a flow-based image animator which enables motion transfer from driving video to target image. An important property of proposed framework is that it supports disentagnlment of motion and appearance. Appearance is represented by the input image, and motion is represented by the generated flow maps which are controlled by the motion code in latent space. Our framework is able to produce the same input image performing diverse motion.  Since motion is only a per-frame 1D vector, one major advantage of our work compared to Video-LDM is the low cost of autoressively generating long videos (e.g., ~1 min).
>
> **Q3. I am missing a section on the limitations and ethical considerations.**
>
> **A3.** We have included the discussion of limitations in Appendix A.4, as well as the Section Ethics Statement in the revised manuscript.

---

### Author Response · Authors · 2023-11-20
**To all reviewers**

We really appreciate three reviewers for their careful reviews and valuable comments. We have uploaded results in the updated Supplementary Material towards better answering reviewers' questions and demonstrating our results.
- Comparison with MoStGAN-V
- Comparison with LVDM
- Comparison with VideoFusion
- Comparison with StyleTalk, ControlVideo and Text2Video-Zero
- Failure cases
- Comparison with Siarohin et al. 2019; 2021

We have uploaded a revised manuscript incorporating reviewers' feedback. Below is a summary of the main changes:
- Missing references are included in Sec. Related Works
- Quantitative comparison with MoStGAN-V is included in Tab. 1
- Ethics Statement
- FVD of unconditional video generation on UCF101 is included in Appendix A.2
- Limitations and Future work are included in Appendix A.4

---

### Meta-Review · Area_Chair_ex6Q · 2023-12-09

**Metareview:**

Paper addresses the task of coherent video generation. Approach disentangles appearance and motion (in the form of optical flow). It first animates the first frame and then projects the motion in the latent space where LDM is used to model dynamics.

The paper was reviewed by three reviewers with mixed scores. Paper received: 2 x Marginally above acceptance and 1 x Reject, not good enough ratings. [8SWB] and [H4YQ] appreciate the work, but are short of actively championing the paper. [NikU] on the other hand raises issues with experimental comparisons (that are lacking) and questions performance on simpler datasets. Notably, [8SWB] is also less confident than the other two reviewers.

AC has read the reviews, rebuttal and discussion (including response from the authors). While AC agrees with some of the author responses, he does not agree with the general arguments that (a) if the model is not officially open-sourced it should not, or cannot, be compared against, and (b) that models that are trained or pre-trained at scale should not be compared against because current model is not trained in this way. In the case of VideoFusion the comparison is fully justified as [NikU] mentions. This is for two reasons: (i) it is published at CVPR'23 which pre-dates submission to ICLR by some time; (ii) they report performance directly on TaiChi dataset, so open-source implementation (or any implantation) should not be necessary for comparison. In opinion of AC authors have no grounds for not comparing to VideoFusion. Authors are correct, however, that LVDM is unpublished and, therefore, comparison to that work is not required.

Overall, this is a borderline paper. Given the lack of championing by any of the reviewers, somewhat lacking experimental comparisons, and, while state-of-the-art, somewhat underwhelming qualitative results the decision is to Reject the paper at this time, giving the opportunity for the authors to address the comments and improve it for resubmission.

**Justification For Why Not Higher Score:**

The qualitative results appear to be somewhat underwhelming given the state-of-art and speed at which generative models are improving. The idea of using the optical flow is also not new. Finally, some of the arguments that authors make in the rebuttal are both a tad to aggressive and at the same time dismissive. Just because certain approaches do not have public implementations, or are trained on more data, does not mean they should be ignored or out of bounds for comparisons.

**Justification For Why Not Lower Score:**

N/A

---

### Decision · Program_Chairs · 2024-01-16

Reject